# Effect of land-use practices on species diversity and selected soil property in Somodo Watershed South-Western Ethiopia

Leta Hailu[1]*, Gizaw Tesfaye[2], Kalkidan Fikirie[3], Yalemtsehay Debebe[1]

1 Jimma Agricultural Research Center, Jimma, Ethiopia, 2 Melkasa Agricultural Research Center, Adama, Ethiopia, 3 Holeta Agricultural Research Center, Holeta, Ethiopia

* latahailu@gmail.com

**Data Availability Statement:** All relevant data are within the manuscript and in supporting information files.

**Funding:** The study was supported by Ethiopian Institute of Agricultural Research.

## Abstract

This study was conducted in Somodo Watershed to investigate the land-use practices and its effect on species diversity and selected soil properties. Field observation was carried out to identify existing land-use practices following a transect line. A total of 20 plots (10 × 10) m² were sampled from plots exhibiting different land-use practices found in the watershed in order to evaluate species richness and diversity. Soil samples were also collected from each plot. The soil samples were analyzed following standard laboratory procedures. The result of the analysis showed that there was a significant difference (p<0.05) in species diversity and richness among different land-use practices. Coffea arabica was dominant in homestead gardens and natural forests while Grevillea robusta showed had maximum richness in plantations and farm forests in the Watershed. Furthermore, home garden agroforestry practice was significantly (p<0.05) affected soil pH compared to other land-use systems (cultivated land, natural forest, and plantation forest. While Organic carbon (OC), Total nitrogen (TN), and Carbon to Nitrogen ratio (C: N) did not show significance difference among land-use systems in the watershed. The study has concluded that different land-use practices had a positive impact on sustaining species diversity, richness, and improve soil properties. Therefore, the study suggests that improving and expanding home garden agroforestry practices in the area are indispensable for environmental protection and soil fertility enhancement.

## Introduction

Land degradation is one of the major barriers to livelihood enhancement in the East Africa Region. In sub-Saharan Africa, about 27 percent of the land area is exposed to land degradation [1]. Likewise, in Ethiopia, it is a serious problem, particularly in the highland areas and generally in the country. The main drivers are deforestation, overgrazing, cultivation of marginal lands, population growth, topography, soil types, agroecology of the area, and insufficient investment in soil and water conservation practices [2, 3] that leads to long-term loss of ecosystem function and productivity of the land [4].

According to [5] greater than 85% of the land is expected to be degraded to various extents. This aggravates the loss of the agricultural productivity and production of the country [6].

**Competing interests:** The authors have declared that no competing interests exist.

This implies that the reversal of land degradation through better understanding of local proximate and underlying drivers [5]. Additionally, certain land-use practices are contributing to biological diversity and soil fertility improvement differently [7, 8]. For instance, the Agroforestry system has the potential to restore degraded land. Agroforestry practice improves and sustains the productive capacity of the land through increasing input of soil organic matter, in the form of surface litter or soil carbon. The practice support livelihoods, improve food security, enhance ecosystem services, and reduce deforestation [9–12].

The climate, soil type, and landforms are determining the type of certain land-use practices in the specific area. Similarly, the land-use practiced also determines the species diversity, richness, and soil properties of that specific area. The agricultural land, forest land, grazing land, and agroforestry practices are the major land-uses practiced predominantly in Somodo Watershed [13]. Several studies were conducted separately on species diversity of different agroforestry practices, the effect of trees on soil properties, and the effect of various land-uses on soil properties across diverse agro-ecologies of the country. For example, the effect of home garden agroforestry on species diversity [14, 15] and the impact of land-use on soil chemical properties [16] were studied. However, the effect of different land-use practices on species diversity and selected soil properties is not yet studied in the watershed. Therefore, the main aim of the study was to investigate the effect of land-use practices in the Somodo Watershed on species richness and diversity, and selected soil physicochemical properties to promote the best fit land-use practice.

## Materials and methods

### Description of the study area

**Location.**    The study was conducted in Somodo Watershed, which is situated in the Abay/Blue Nile river basin, in the upper part of Dhidhesa catchment, Jimma Zone Oromia Regional State, Ethiopia. It is located about 15Km West of Jimma town and 368Km Southwest of Addis Ababa the capital city of Ethiopia. Somodo Watershed covers about 400 hectares of land and comprises 300 households. The Watershed is geographically located between 7°45"30'N-7°47"00'N latitude and 36°48"00'E-36°49"00'E longitude (Fig 1).

**Vegetation.**    Several native and exotic plant species commonly found in most part of the area. For instance, natural forest dominated by *Albizia gummifera* and *Millettia ferruginea* while home garden agroforestry was dominated by *Coffea arabica*, *Persea americana*, *Psidium guajava* and *Cordia africana* tree species. On the other hand, in farm forestry *Coffea arabica*, *Grevillea robusta* and *Croton macrostachyus* species are commonly found and plantation forestry was controlled through *Grevillea robusta* and *Coffea arabica* species.

**Agricultural activities.**    The agricultural activities of the watershed are mainly characterized by the existence of subsistence mixed farming of both agricultural crop production and livestock. The major cereal crops produced in the watershed are maize (*Zea mays*), teff (*Eragrostiestefzucca*), soybean, wheat *(Triticumaestivium)*, barley (*Hordiumvulgarae*), sorghum (*Sorghum bicolor*), and cash crop like coffee (*Coffea arabica*), khat (Catha edulis), and fruit (Mango, Avocado, papaya). The community used fuelwood as a source of energy. Their main source of income is obtained from the sale of crops, fruits, and livestock [19].

### Research design and soil sampling

Field observation was carried out to identify and characterize the existing land-use practices. Purposive sampling technique was used for selecting on-farm study plots where land-uses are practiced. Field observations were held along a transect line with plots measuring $(10 \times 10)$ m$^2$, was established along the transect line as the land-uses appear. A total of 20 plots were sampled

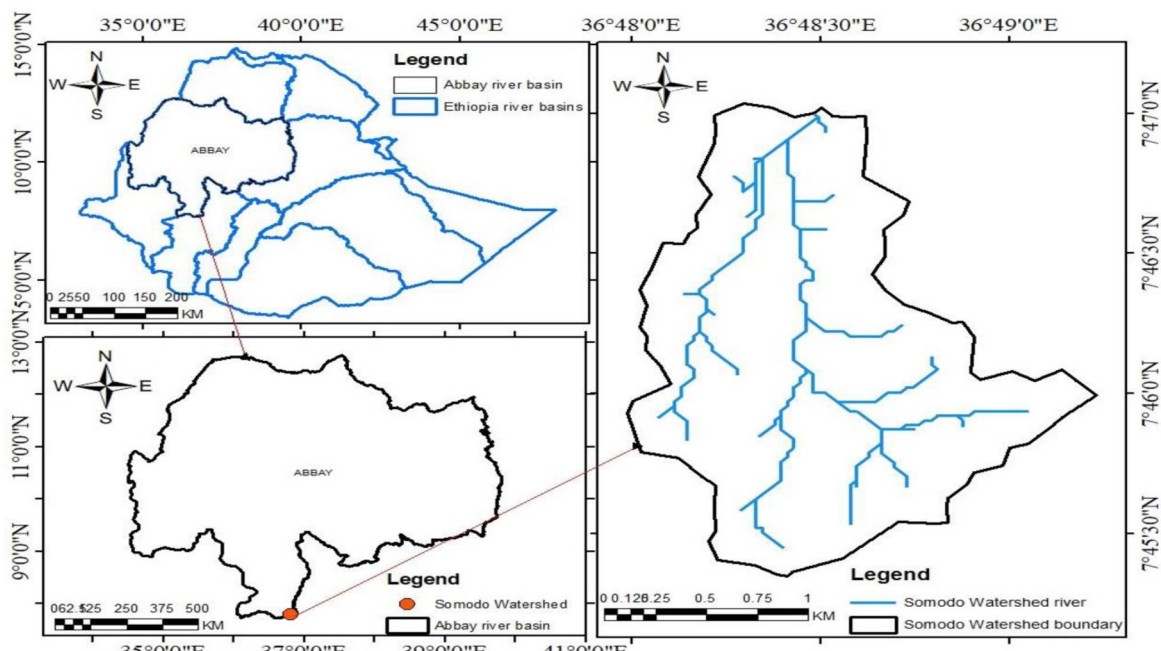

**Fig 1. Map of Somodo watershed, Jimma zone, southwestern Ethiopia.** Agroecology and soil: Somodo Watershed is characterized by the Agroecological zone of Tepid sub-humid and mid-highland (SH3). The altitude of the study area ranges from 1900 to 2075 m.a.s.l. The long-term means annual rainfall of the watershed is 1954.3 mm with the mean temperature of 19.3°C ranging from 15.0 T min to 23.5 T max (Fig 2). Nitisols and Orthic Acrisols are the most dominant soil types of the area and Nitisols accounting for 64% of slightly acidic soil [17]. The textural class of the soil is sandy clay loam, while the pH is found to be strongly acidic [18].

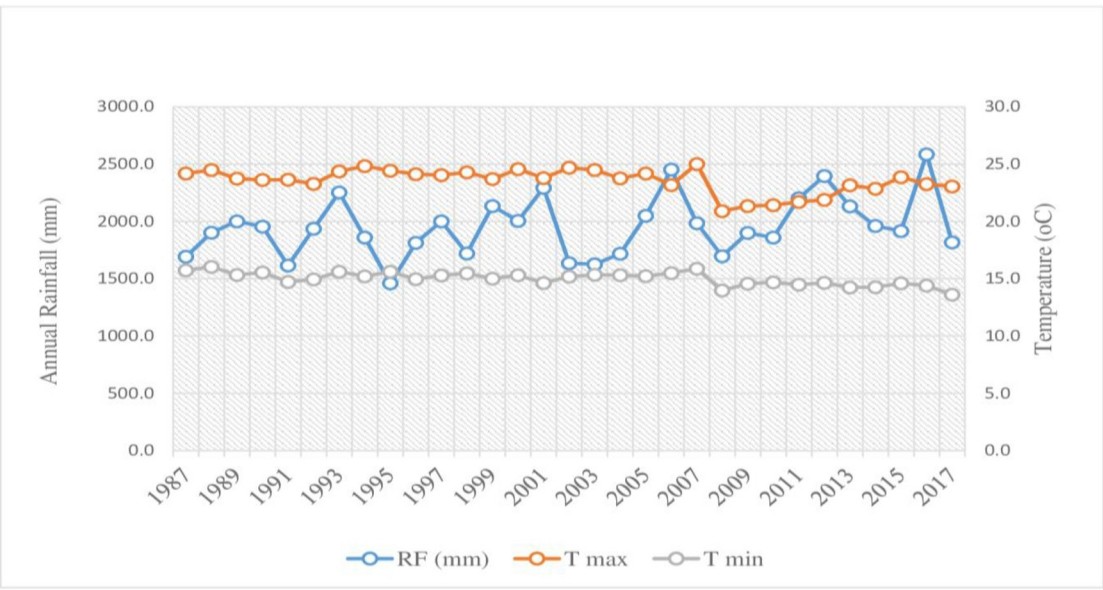

**Fig 2. Mean annual RF, max and min temperature of Somodo watershed.** Land-use: Somodo watershed is characterized by a coffee-based farming system and agroforestry practices. Land-use of the watershed described as cultivation land (land covered by annual crop), Forest land (natural and plantation), grazing land, and agroforestry (home garden) are the major land-uses. The dominant land-uses are agroforestry practice (46.97%), cultivation land (21.26%), forest land (18.51%), and grazing land (13.26%), respectively [16].

from different land-use practices. In each plot, all woody species were identified by their local and/or scientific names, and identities that help for identification was recorded to produce a more complete list of the woody plants in the study area. The environmental variables, viz altitude, and coordinates of each plot were taken with GPS (Geographical positioning system). For each type of land-use composite soil samples were collected from a depth 0–30 cm by using sharp-edged, closed, and circular auger pushed manually down the soil profile. The soil samples were collected on August 18th, 2016. The collected soil samples were transported to the Jimma Agricultural Research Centre Soil, water, and plant tissue laboratory for analysis and determination of soil property variation.

## Laboratory analyses

pH of soil was determined by a water suspension method with a microprocessor-based pH system on a 1:2.5 soil to water ratio [20]. Organic carbon (OC) was determined by Walkley and black method [21]. Kjeldahl digestion, distillation, and titration method [22] were used for total N (TN) determination. Bray II extraction method with spectrophotometer analysis was used for available phosphorus (Av.-P) determination [23].

## Data analysis

Importance Value Index (IVI) was computed using the following formula [24].

$$IVI = RD + RF + RM \tag{1}$$

Where RD is relative density; RF is the relative frequency and RM is relative dominance. The collected qualitative and quantitative data were analyzed using SPSS Version 20. The frequency of tree species was described using bar graphs with the help of Microsoft Excel 2010. Soil data were tested using ANOVA following the generalized linear model (GLM) procedure at 5% level of probability using the Statistical software for social science (SPSS) [25]. List significance difference (LSD) was used to separate means of treatments when they are significantly different ($p \leq 0.05$).

## Results and discussion

### Characterization and species richness of land-use practices

Land-use practices found in the watershed were categorized into the home garden, natural forest, plantation forestry, and farm forestry (Fig 3). Consequently, diversified species were observed in the home garden, natural forest, plantation forest, and farm forest, respectively. In-home garden agroforestry *Coffea arabica*, *Persea americana*, *Psidium guajava*, and *Cordia africana* are the most dominant species. According to [8], home garden agroforestry was more diverse and richer in species than parkland agroforestry. A similar finding was also reported by [14] home garden agroforestry had diversified species in the study conducted in Jabithenan district, Northwestern Ethiopia. While in plantation land-use, *Grevillea robusta* and *Coffea arabica* had the highest frequencies in the watershed (Fig 3). On the other hand, *Coffea arabica*, *Albizia gummifera*, and *Millettia ferruginea* are dominant species in natural forests.

Regarding the importance value index (IVI), *Coffea arabica* (103%) and *Albizia gummifera* (47.77%) were higher importance value index in home garden agroforestry. In plantation forests, *Coffea arabica* (61.3%) and *Croton macrostachyus* (59.5%) have great IVI in the Somodo watershed. From natural forest and farm forest, *Coffea arabica* (108.2%), *Grevillea robusta* (104.99%) and *Croton macrostachyus* (48%) have higher IVI, respectively. IVI value is an important parameter that reveals the ecological significance of species in a given ecosystem.

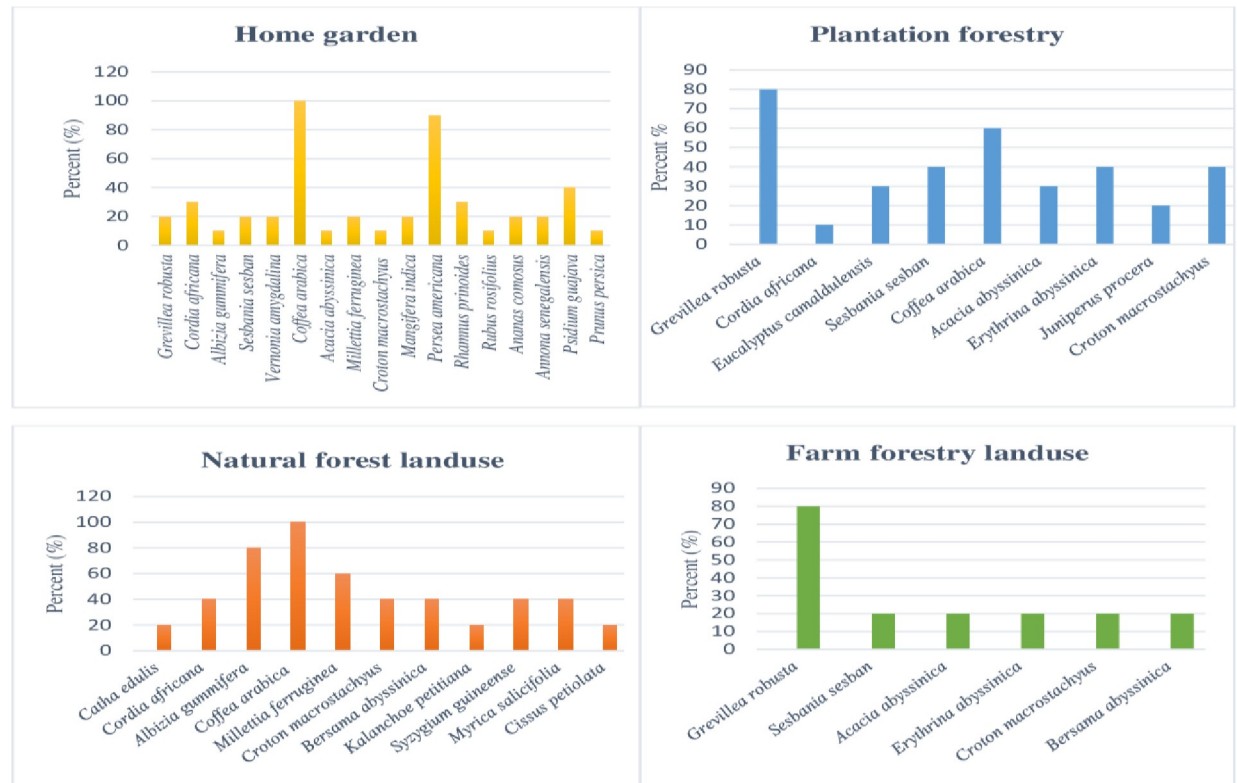

**Fig 3. Home garden, plantation, natural forest and farm forestry land use species diversity in the Somodo Watershed.**

Therefore, *Coffea arabica*, *Albizia gummifera*, *Grevillea robusta*, and *Croton macrostachyus* are widely adapted and economically important in the study area.

## Effect of land-use practices on selected soil chemical properties

Land-use practices affect soil physicochemical and biological properties differently. The higher mean of soil pH (5.41) was recorded under the home garden agroforestry compared to other land-use types (Table 1). This shows that soil found under the home garden was less acidic than a plantation, natural forest, and cultivated land-use types. Perhaps due to the accumulation of forest litter from diverse species and input of the waste product from the house; that

**Table 1. Mean value (±SDM) of soil pH, available phosphorous, organic carbon, total N, and carbon to nitrogen ratio with different land-use.**

| Land-use types | Soil parameters | | | | |
|---|---|---|---|---|---|
| | pH (H₂O) | Av.P (ppm) | OC (%) | TN (%) | C:N |
| Plantation | 4.91±0.24 | 3.49±2.40 | 2.33±0.57 | 0.23±0.06 | 10.54±2.29 |
| Forest | 4.89±0.30 | 2.65±2.24 | 2.45±0.35 | 0.23±0.05 | 11.14±1.75 |
| Cultivation | 4.90±0.30 | 1.94±0.60 | 2.78±0.41 | 0.31±0.06 | 9.1±1.09 |
| Home garden | 5.41±0.14 | 4.00±1.11 | 2.85±0.30 | 0.25±0.03 | 8.93±5.23 |
| CV (%) | 4.27 | 57.78 | 12.25 | 23.8 | 32.85 |
| LSD | 0.296 | 2.405 | 0.44 | 0.08 | 4.49 |

*Note*: *pH*: *soil pH; Av.-P: Available phosphorus; OC: Organic carbon; TN: Total nitrogen; CV: Coefficient of variation; LSD: List significant difference; SDM: Standard deviation of Mean*

improves the soil organic carbon of soil under the home garden and ameliorated the soil pH. Similar result was reported by [26], soil pH under the home garden was significantly higher than other land-use systems in Gununo watershed.

Regarding phosphorus, there was no significant difference (p<0.05) observed between plantation, natural forest, and home garden agroforestry systems. But, the higher mean of available phosphorus was observed under the home garden compared to other land-use practices. This is perhaps due to the strong soil acidity and inherent properties of the soil in the study area. Phosphorus is one of the essential elements required by the plant next to nitrogen to complete its life cycle than other essential elements. Hence, soil organic matter management may ameliorate soil pH and improves the availability of phosphorus.

Soil organic carbon (SOC) did not show a significant difference (P<0.05) under different land-use practices in Somodo watershed. Nevertheless, the higher mean was recorded under home garden agroforestry practices than the other land-use practices (Table 1). This may be due to the removal of biomass and crop residue for animal feeding and using as energy sources. According to [27] organic matter (OM) has an important influence on soil physical and chemical characteristics, soil fertility status, plant nutrition, and biological activity in the soil. Therefore, the management of soil organic matter is crucial to maintain the healthy functioning of the soil system.

Total nitrogen (TN) was not exhibited a significant difference (P<0.05) among various land-use practices in the watershed (Table 1). However, a higher mean was observed under cultivation land. This is possible due to the input of inorganic fertilizer on cultivated land; TN measures the total amount of nitrogen present in the soil including both organic and inorganic forms [28]. On the contrary, another study elsewhere confirmed that total nitrogen was significantly different for topsoil under home garden agroforestry practice than other land-use practices [29]. The carbon to nitrogen ratio also was not shown a significant difference (P<0.05) between the land-uses in the watershed. However, a higher mean of C: N was observed under forest land (11:1) than the other land-use practices. This might be due to the low input of the organic matter under the land-uses. This requires maintenance of the soil carbon stock.

## The implication of the study

The implication of this study was different land-use practices affect the species diversity and richness in the watershed. For instance, *Coffea arabica* was the dominant one since the area is the coffee growing area. Compared to this, *Grevillea robusta* species is the popular species in the area since farmers used for farm boundary plantation and around homestead. Furthermore, land-use practices affect soil properties positively and negatively. Hence, improving and expanding agroforestry practices are vital for the improvement of environmental protection.

## Conclusion

The result of the analysis revealed that diversified species were observed in home garden agroforestry practice than the other land-use practices. *Coffea arabica*, *Albizia gummifera*, *Grevillea robusta*, and *Croton macrostachyus* were widely adapted species and economically very important in the watershed. Regarding soil property improvement, home garden agroforestry practice was significantly (p<0.05) affected soil pH compared to cultivated land, natural forest, and plantation forest land-use systems. On the other hand, organic carbon, total nitrogen, and carbon to nitrogen ratio did not show significant differences under different land-use practices in the study area. Therefore, home garden agroforestry practices are a very important land-use system in Somodo watershed that contributes to local livelihood improvement. This study

suggest agroforestry practice expansion in the watershed through participatory awareness creation with all stakeholders is essential on its environmental and economic importance.

## Supporting information

**S1 Data.**
(XLSX)

## Author Contributions

**Data curation:** Leta Hailu.

**Formal analysis:** Kalkidan Fikirie.

**Investigation:** Gizaw Tesfaye, Kalkidan Fikirie, Yalemtsehay Debebe.

**Writing – original draft:** Leta Hailu, Kalkidan Fikirie.

**Writing – review & editing:** Leta Hailu, Gizaw Tesfaye, Kalkidan Fikirie, Yalemtsehay Debebe.

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
