## [Editor Report · Decision Letter 0]

18 Nov 2020

PONE-D-20-33362

Effect of Land-use Practices on Species Diversity and Selected Soil Property in Somodo Watershed South-Western Ethiopia

PLOS ONE

Dear Dr. Hailu,

Thank you for submitting your manuscript to PLOS ONE. After careful consideration, we feel that it has merit but does not fully meet PLOS ONE’s publication criteria as it currently stands. Therefore, we invite you to submit a revised version of the manuscript that addresses the points raised during the review process.

Please incorporate the followings before I invite reviewers on the revised version of your manuscript

(i) Provide the site characteristics including slope, aspect, vegetation characteristics for each of the land-uses selected for the study.

(ii) Provide the original reference for the IVI

(iii) Correct the spelling of Jackson 1958

(iv) Provide the axis title for the Figure 3.

(v) Add a new section on the implication of the study

We look forward to receiving your revised manuscript.

Kind regards,

Arun Jyoti Nath

Academic Editor

PLOS ONE

Journal Requirements:

3. We note that Figure 1 in your submission contains map images which may be copyrighted.

We require you to either (a) present written permission from the copyright holder to publish these figure specifically under the CC BY 4.0 license, or (b) remove the figure from your submission:

b. If you are unable to obtain permission from the original copyright holder to publish these figure under the CC BY 4.0 license or if the copyright holder’s requirements are incompatible with the CC BY 4.0 license, please either i) remove the figure or ii) supply a replacement figure that complies with the CC BY 4.0 license. Please check copyright information on all replacement figures and update the figure caption with source information. If applicable, please specify in the figure caption text when a figure is similar but not identical to the original image and is therefore for illustrative purposes only.
---

## [Author Response · Author response to Decision Letter 0]

31 Jan 2021

comments and suggestion of the reviewers and editorial is considered in revised version of the manuscript, if any more concerns are present, we are ready to respond. many thanks

---

## [Decision Letter · Decision Letter 1]

17 Feb 2021

PONE-D-20-33362R1

Effect of Land-use Practices on Species Diversity and Selected Soil Property in Somodo Watershed South-Western Ethiopia

PLOS ONE

Dear Dr. Gemechu,

Thank you for submitting your manuscript to PLOS ONE. After careful consideration, we feel that it has merit but does not fully meet PLOS ONE’s publication criteria as it currently stands. Therefore, we invite you to submit a revised version of the manuscript that addresses the points raised during the review process.

ACADEMIC EDITOR:

A major revision has been suggested by the reviewer. Please revise it accordingly.

We look forward to receiving your revised manuscript.

Kind regards,

Arun Jyoti Nath

Academic Editor

PLOS ONE

Reviewers' comments:

Reviewer's Responses to Questions

**Comments to the Author**

1. If the authors have adequately addressed your comments raised in a previous round of review and you feel that this manuscript is now acceptable for publication, you may indicate that here to bypass the “Comments to the Author” section, enter your conflict of interest statement in the “Confidential to Editor” section, and submit your "Accept" recommendation.

Reviewer #1: (No Response)

2. Is the manuscript technically sound, and do the data support the conclusions?

Reviewer #1: Partly

3. Has the statistical analysis been performed appropriately and rigorously? 

Reviewer #1: Yes

4. Have the authors made all data underlying the findings in their manuscript fully available?

Reviewer #1: Yes

5. Is the manuscript presented in an intelligible fashion and written in standard English?

Reviewer #1: No

6. Review Comments to the Author

Reviewer #1: The manuscript titled “Effect of Land-use Practices on Species Diversity and Selected Soil Property in Somodo Watershed South-Western Ethiopia” was reviewed and found that the authors have not written it with clarity. Grammatical and typographical errors are numerous to list. Scientific names of plant species need to written in italics. The manuscript is not suitable for publication in its current form. Authors are advised to revised it critically and resubmit. The following observations may also be addressed adequately while revising the manuscript.

P1 L10: “the” land-use practices.

P1 L12: ‘trail walk’ or ‘transect line’ instead of ‘transect walk’, (10 ×10) m2.

P1 L13, 14 & 15: “were sampled from plots exhibiting different land-use practices found in the watershed in order to evaluate species richness and diversity. Soil samples were also collected from each plot. The soil samples were analysed following standard laboratory procedures.

P1 L15:shows showed.

P1 L16:between among.

P1 L17 & 18: Coffea arabica was dominant in homestead gardens and natural forests while Giravilia robusta showed had maximum richness in plantations and farm forests in Somodo Waatershed.

P1 L19-21: There is a lack of clarity in interpretation and presentation of major findings of the study. Abbreviations need to be explained and results must appear before moving on to deductions. Please write full form of OC, TN and C:N in the abstract or in its first appearance. The abstract is completely descriptive, it requires to be written highlighting the key findings on species diversity, richness and soil parameters analysed in the study.

Comment on L 10-25: The abstract requires a grammatical check. Proper data-oriented results should lead to deductions and conclusions.

P2 L 26:several degrees various extents.

P2 L37:

• exacerbates aggravates

• This implies that the reversal of land degradation through better understanding of local proximate and underlying drivers.

P2 L40: to different degrees.

P2 L36-44: Implications of words like “that”, “these” are not clear. It shall be better if it is replaced by scientific terminologies or methodologies as the case may be.

P3 L 61: “in” the upper part of Dhidhesa catchment.

P3 L 72:account accounting.

P4 L 73 & 74:and the pH of the soil is rated as strong acidity. while the pH is found to be strongly acidic.

Figure 2: The content of the ‘description of study area’ do not reveal clear correlation between rainfall and temperature. Therefore, it is advised to use two separate frequency line graphs depicting patterns of annual rainfall and temperature (max & min).

P4 L 77 & 78: What is meant by “and agro-forestry practices”? It needs to be clarified in context to the prevailing land use practice.

P4 L 82: Capitalize ‘several’ to ‘Several’.

P4 L82-87: The paragraph requires careful grammatical check. The words ‘dominated’ and ‘conquered’ have been interchangeably used, which makes it unclear. Use of proper punctuations at the correct places is advised, the lack of which renders the sentence a different meaning.

P5 L 90:Maize maize.

P5 L94 & 95:their main source of income is gained from the selling of their main source of income is obtained from the sale of crops, fruits and livestock. The use of coordinating conjunctions is not clear in this paragraph.

P5 L 96-109:

• Field observations were held along a transect line with plots measuring (10 × 10) m2. was established along the transect line as the land-uses appear.

• Namely should be replaced by ‘viz.’

• Soil sampling technique using augar should follow some protocol.

P5 L 111: Since the term, ‘soil reaction’ has not been used before this; it is advisable to write it as ‘pH of soil’.

P6 L 114: This line mentions processes which require citation.

P5 L119: The use of “the” needs to be checked.

P5 L 121:‘Microsoft’ Excel ‘version?’.

P5 L122: General linier model generalised linier model.

P5 L 124: What is meant by “for windows”?

P5 L124: LSD is Least significant difference not list significant difference.

P5 L125: when they are significantly different (p≤0.05).

P 5 L 128:characterized categorized.

P 5 & 6 L 127-137:

• Scientific names need to be written in italics following taxonomic norms.

• Use of proper punctuations is required.

P 6 L 138-145:

• Proper framing of sentences coupled with the use of proper punctuations and a grammatical check is suggested.

• Also, look into the plagiarism of this paragraph.

• Singular and plural forms should be properly deployed.

Figure 3 should be better represented with a pie chart.

P 8 L 149-157:

• Grammatical check for this paragraph is suggested.

• Comparisons from figure 3 are not clear. Thus, proper use of wordings and punctuations are suggested.

P 8 L 158 – 164:

• Please mention the significance level of testing in line number 158-159.

• Use of proper punctuations is desirable in case of comparison statements.

P 8 L 165: Kindly mention the significance level for hypothesis testing.

P 9 L 173 & 174: Total Nitrogen (T.N.) did not exhibit significance difference (at level of significance?) among various land-use practices in the study watershed.

Table 1: Please maintain uniformity in alignment of data in table.

P 9 L 190:coffee ArabicaCoffea arabica.

P 9 L 191:Concerning this In contrast/ Compared to this.

P 10 L 197 – 198: Please write scientific names in italics following taxonomic norms.

P 10 L 204: The study suggests This study suggest.

7. PLOS authors have the option to publish the peer review history of their article (what does this mean?). If published, this will include your full peer review and any attached files.

Reviewer #1: No

---

## [Author Response · Author response to Decision Letter 1]

16 Mar 2021

1. P1 L10: “the” land-use practices, thank you for your suggestion, now corrected with appropriate punctuation (the land-use practice)

2. P1 L12: ‘trail walk’ or ‘transect line’ instead of ‘transect walk’, (10 ×10) m2. Thank you, now corrected to transect line & (10 ×10) m2

3. P1 L13, 14 & 15: “were sampled from plots exhibiting different land-use practices found in the watershed in order to evaluate species richness and diversity. Soil samples were also collected from each plot. The soil samples were analyzed following standard laboratory procedures. Thank you for your suggestion and the comments considered.

4. P1 L15: shows showed. Corrected.

5. P1 L16: between among. Now corrected.

6. P1 L17 & 18: Coffea arabica was dominant in homestead gardens and natural forests while Giravilia robusta showed had maximum richness in plantations and farm forests in Somodo Waatershed. Now corrected accordingly.

7. P1 L19-21: There is a lack of clarity in interpretation and presentation of major findings of the study. Abbreviations need to be explained and results must appear before moving on to deductions. Please write full form of OC, TN and C: N in the abstract or in its first appearance. The abstract is completely descriptive, it requires to be written highlighting the key findings on species diversity, richness and soil parameters analysed in the study. Thank you for your constructive comments and now improved accordingly. 

8. Comment on L 10-25: The abstract requires a grammatical check. Proper data-oriented results should lead to deductions and conclusions. Thank you for your comment, now tried to improve

9. P2 L 26: several degrees various extents. Now replaced with the suggested one 

10. P2 L37:

• exacerbates aggravates

• This implies that the reversal of land degradation through better understanding of local proximate and underlying drivers. Now updated with the suggested one

11. P2 L40: to different degrees. Now corrected

12. P2 L36-44: Implications of words like “that”, “these” are not clear. It shall be better if it is replaced by scientific terminologies or methodologies as the case may be. Thank you for the comment, now replaced with the correct one

13. P3 L 61: “in” the upper part of Dhidhesa catchment. Considered.

14. P3 L 72: account accounting. Now updated with suggested one.

15. P4 L 73 & 74:and the pH of the soil is rated as strong acidity. while the pH is found to be strongly acidic. Now corrected with the suggested one.

16. Figure 2: The content of the ‘description of study area’ do not reveal clear correlation between rainfall and temperature. Therefore, it is advised to use two separate frequency line graphs depicting patterns of annual rainfall and temperature (max & min). Thank you for the suggestion, here, the objective is to show the trend and long term mean of rain fall, min & max temperature of the area.

17. P4 L 77 & 78: What is meant by “and agro-forestry practices”? It needs to be clarified in context to the prevailing land use practice. Now corrected to home garden agroforestry which is stands for the prevailing land-use in the study.

18. P4 L 82: Capitalize ‘several’ to ‘Several’. Now corrected to the suggested one.

19. P4 L82-87: The paragraph requires careful grammatical check. The words ‘dominated’ and ‘conquered’ have been interchangeably used, which makes it unclear. Use of proper punctuations at the correct places is advised, the lack of which renders the sentence a different meaning. Sure, the comments are considered and corrected accordingly.

20. P5 L 90: Maize maize. Now corrected

21. P5 L94 & 95: their main source of income is gained from the selling of their main source of income is obtained from the sale of crops, fruits and livestock. The use of coordinating conjunctions is not clear in this paragraph. Thank you for the comment, now corrected according to the suggested one.

22. P5 L 96-109:

• Field observations were held along a transect line with plots measuring (10 × 10) m2. was established along the transect line as the land-uses appear. Now corrected.

• Namely should be replaced by ‘viz.’ Now corrected.

• Soil sampling technique using augar should follow some protocol. Thank you for the comment soil sample collection using auger is possible up to 0 to 30 cm. 

23. P5 L 111: Since the term, ‘soil reaction’ has not been used before this; it is advisable to write it as ‘pH of soil’. Now corrected.

24. P6 L 114: This line mentions processes which require citation. Thank you for the comment, I think the citation is already there.

25. P5 L119: The use of “the” needs to be checked. Considered 

26. P5 L 121: ‘Microsoft’ Excel ‘version?’. Thank you, now corrected.

27. P5 L122: General linier model generalized linier model. Thank you, corrected.

28. P5 L 124: What is meant by “for windows”? 

29. P5 L124: LSD is Least significant difference not list significant difference. Now Corrected.

30. P5 L125: when they are significantly different (p≤0.05). corrected.

31. P 5 L 128: characterized categorized.

32. P 5 & 6 L 127-137:

• Scientific names need to be written in italics following taxonomic norms.

• Use of proper punctuations is required. Thank you for the comment, the suggestion is considered.

33. P 6 L 138-145:

• Proper framing of sentences coupled with the use of proper punctuations and a grammatical check is suggested. Sure, considered.

• Also, look into the plagiarism of this paragraph.

• Singular and plural forms should be properly deployed. Sure, considered.

Figure 3 should be better represented with a pie chart. Thank you for your suggestion. 

34. P 8 L 149-157:

• Grammatical check for this paragraph is suggested.

• Comparisons from figure 3 are not clear. Thus, proper use of wordings and punctuations are suggested. Thank for the suggestions, now corrected.

35. P 8 L 158 – 164:

• Please mention the significance level of testing in line number 158-159. Thank you for the comment, now mentioned.

• Use of proper punctuations is desirable in case of comparison statements. Thank you, now corrected.

36. P 8 L 165: Kindly mention the significance level for hypothesis testing.

37. P 9 L 173 & 174: Total Nitrogen (T.N.) did not exhibit significance difference (at level of significance?) among various land-use practices in the study watershed. Now the significance level is mentioned and the suggestion is accepted accordingly.

Table 1: Please maintain uniformity in alignment of data in table. Now the uniformity of the data in the table is corrected.

38. P 9 L 190: coffee Arabica Coffea arabica. Now corrected.

39. P 9 L 191: Concerning this In contrast/ Compared to this. Now updated with the suggested one.

40. P 10 L 197 – 198: Please write scientific names in italics following taxonomic norms. Now corrected.

41. P 10 L 204: The study suggests This study suggest. Corrected accordingly.

---

## [Decision Letter · Decision Letter 2]

19 Apr 2021

PONE-D-20-33362R2

Effect of Land-use Practices on Species Diversity and Selected Soil Property in Somodo Watershed South-Western Ethiopia

PLOS ONE

Dear Dr. Gemechu,

Thank you for submitting your manuscript to PLOS ONE. After careful consideration, we feel that it has merit but does not fully meet PLOS ONE’s publication criteria as it currently stands. Therefore, we invite you to submit a revised version of the manuscript that addresses the points raised during the review process.

ACADEMIC EDITOR: Please consider the reviewer comments and revise the manuscript accordingly

We look forward to receiving your revised manuscript.

Kind regards,

Arun Jyoti Nath

Academic Editor

PLOS ONE

Journal Requirements:

Reviewers' comments:

Reviewer's Responses to Questions

**Comments to the Author**

1. If the authors have adequately addressed your comments raised in a previous round of review and you feel that this manuscript is now acceptable for publication, you may indicate that here to bypass the “Comments to the Author” section, enter your conflict of interest statement in the “Confidential to Editor” section, and submit your "Accept" recommendation.

Reviewer #1: All comments have been addressed

2. Is the manuscript technically sound, and do the data support the conclusions?

Reviewer #1: Yes

3. Has the statistical analysis been performed appropriately and rigorously? 

Reviewer #1: Yes

4. Have the authors made all data underlying the findings in their manuscript fully available?

Reviewer #1: Yes

5. Is the manuscript presented in an intelligible fashion and written in standard English?

Reviewer #1: Yes

6. Review Comments to the Author

Reviewer #1: The revised version of manuscript ID PONE-D-20-33362R2 was reviewed and found that authors have improved it upto a significant extent possible. Authors are advised to correct scientific name of plant species in Figure 3 as well. All names must be written in italic.

7. PLOS authors have the option to publish the peer review history of their article (what does this mean?). If published, this will include your full peer review and any attached files.

Reviewer #1: **Yes: **Dr. Krishna Giri

---

## [Editor Report · Decision Letter 3]

14 May 2021

Effect of Land-use Practices on Species Diversity and Selected Soil Property in Somodo Watershed South-Western Ethiopia

PONE-D-20-33362R3

Dear Dr. Gemechu,

We’re pleased to inform you that your manuscript has been judged scientifically suitable for publication and will be formally accepted for publication once it meets all outstanding technical requirements.

Kind regards,

Arun Jyoti Nath

Academic Editor

PLOS ONE
---

## [Editor Report · Acceptance letter]

19 May 2021

PONE-D-20-33362R3 

Effect of Land-use Practices on Species Diversity and Selected Soil Property in Somodo Watershed South-Western Ethiopia 

Dear Dr. Hailu:

I'm pleased to inform you that your manuscript has been deemed suitable for publication in PLOS ONE. Congratulations! Your manuscript is now with our production department. 

Kind regards, 

on behalf of

Dr. Arun Jyoti Nath 

Academic Editor

PLOS ONE